# Frailty Status and Polypharmacy Predict All-Cause Mortality in Community Dwelling Older Adults in Europe

**DOI:** 10.3390/ijerph18073580

**Published:** 2021-03-30

**Authors:** Luís Midão, Pedro Brochado, Marta Almada, Mafalda Duarte, Constança Paúl, Elísio Costa

**Affiliations:** 1UCIBIO REQUIMTE, Faculty of Pharmacy, Porto4Ageing, University of Porto, 4050-313 Porto, Portugal; luismidao@gmail.com (L.M.); pedrombpb@gmail.com (P.B.); martassalmada@gmail.com (M.A.); 2Institute of Biomedical Sciences Abel Salazar, University of Porto, 4050-313 Porto, Portugal; paul@icbas.up.pt; 3CINTESIS—Center for Health Technology and Services Research, 4200-450 Porto, Portugal; mafaldaduarte@hotmail.com; 4Higher Education Institute of Health of Alto Ave, 4720-155 Amares, Portugal

**Keywords:** frailty, polypharmacy, all-cause mortality, follow-up, 30 months follow-up

## Abstract

European population ageing is associated with frailty, a complex geriatric syndrome, and polypharmacy, both resulting in adverse health outcomes. In this study we aimed to evaluate the impact of frailty and polypharmacy, on mortality rates, within 30 months, using a cohort of SHARE participants aged 65 years old or more. Frailty was assessed using a version of Fried’s phenotype criteria operationalized to SHARE while polypharmacy was defined as taking five or more drugs per day. We found a prevalence of 40.4% non-frail, 47.3% pre-frail and 12.3% frail participants. Moreover, a prevalence of polypharmacy of 31.3% was observed, being 3 three times more prevalent in frail individuals and two times in pre-frail individuals, when compared with non-frail. Individuals with both conditions had shown higher mortality rates. Comparing with non-polymedicated non-frail individuals all the other conditions are more prone to die within 30 months. Polymedicated older and male participants exhibited also higher mortality rates. This work shows polypharmacy and frailty to be associated with a higher risk of all-cause of mortality and highlights the need to decrease ‘unnecessary’ polypharmacy to reduce drug-related issues and also the need to assess frailty early to prevent avoidable adverse outcomes.

## 1. Introduction

The European population has been growing old over the last decades and this is expected to continue, due to better quality of life and increased life expectancy [1,2]. Higher longevity leads to a greater expression of ageing related conditions, such as chronic diseases, comorbidities, and geriatric syndromes, which poses a serious challenge at economic and health care systems levels [3,4].

Frailty, usually characterized by a reduction in physiological reserves and loss of resistance to stressors caused by deficits related to ageing, has been attracting the interest of the scientific community over the past years [5]. This geriatric syndrome represents a potentially huge public health problem due to the clinical and societal consequences of its dynamic nature, affecting not only the individuals but their caregivers, the healthcare system and society [6,7]. There are a lot of instruments to assess frailty, although, the Frailty Phenotype, defined by Fried et al., [8] is the most frequently used instrument in both research and clinical practice [9]. Being (pre-)frail is associated with various poor health outcomes, such as falls, hospitalizations, disability, low quality of life and death [5,8]. Frailty is indeed a strong predictor of all-cause mortality [10,11,12,13].

Due to the increased expression of chronic conditions with ageing, polypharmacy is a highly prevalent condition among the older population aged 65 years old or more [14]. Defined as the use of multiple drugs by a patient, the minimum number of drugs needed to define polypharmacy is variable, usually between 5 to 10. Although polypharmacy often refers to prescription drugs, it is also important to consider over-the-counter drugs, supplements, and other products [15,16]. Regardless of the number of drugs used to define polypharmacy, there was already reported that polymedicated individuals have higher rates of death, being these rates higher with an increasing number of taken drugs (1–4, 5, 6–9 or >10) [17]. Other previous studies on community-dwelling older persons showed an association between polypharmacy and all-cause mortality [18].

The relationship between these two common conditions among the older population, polypharmacy, and frailty, has been previously studied. Some studies showed that indeed polypharmacy is associated with frailty, with frail people having higher rates of polypharmacy [19,20]. Another study showed that older adults who experience polypharmacy had a higher risk of developing frailty within 3 years [19,21]. Another study over 6 years confirmed that polypharmacy significantly increases the risk of frailty in older adults [22]. Although, one of the problems found by a systematic review of the relationship between frailty and polypharmacy was the lack of homogeneity on how frailty and polypharmacy are assessed/measured and quantified [23].

Some studies already reported that polypharmacy among pre-frail and frail individuals is associated with higher rates of mortality in Spain, the United Kingdom and France [24,25,26]. Although, to our knowledge, no study was performed studying the relation between polypharmacy and frailty in mortality, from a European perspective. Therefore, with this study, we aimed to analyse the relationship between frailty and polypharmacy, and its impact on mortality rates, within 30 months, using a cohort of the harmonized European dataset, Survey of Health, Ageing, and Retirement in Europe (SHARE), of older adults aged 65 years old or over.

## 2. Materials and Methods

### 2.1. Study Participants

SHARE, the acronym of Survey of Health, Ageing and Retirement in Europe, is a longitudinal cohort study of community-dwelling people aged 50 years old or over, from 27 European countries plus Israel, that provides information of more than 140,000 individuals. Since its inception in 2004, SHARE has released data every 2 years, having already released seven waves of data. Wave 6 (with data from 2015) and wave 7 (with data from 2017), are the most recent waves. In this work, we used data from wave 6 and 7. From the 68,231 participants included in SHARE’s wave 6, we selected those without missing data for questions related to age, gender, polypharmacy, and frailty status assessment, aged 65 years old or more, and who were followed up until wave 7, or those who died who had information about the month of decease. We extracted the data files including information on age, gender, polypharmacy and frailty, time of death (for those who deceased), and combined them into a single data set.

### 2.2. Frailty

An operationalized version of Fried’s phenotype to SHARE was used to assess frailty. Frailty status was defined as previously described [27] and that is based on five dimensions: exhaustion, shrinking, weakness, slowness, and low activity.

For each fulfilled criterion, one point was assigned to each participant. Individuals with zero points were classified as robust, those with 1 or 2 points as pre-frail and those with 3 to 5 points were classified as frail.

-One point was assigned for those with a positive answer to “In the last month, have you had too little energy to do the things you wanted to do?” fulfilling the exhaustion criterion.-Those that reported “diminution in desire for food” to “What has your appetite been like in the last month?” or, in the case of an encodable answer to this question, those that reported “less” to “So, have you been eating more or less than usual?”, had one point in the shrinking criterion.-Weakness criteria, derived from the highest of 4 handgrip strength measurements, adjusted by gender and body mass index (BMI), and was fulfilled by men with handgrip strength ≤ 29, ≤30 or ≤32, if BMI ≤ 24, 24 < BMI ≤28 or ≤ 32, respectively; and by women with handgrip strength ≤ 17, ≤ 17.3, ≤ 18 or ≤ 21 if BMI ≤ 23, 23 < BMI ≤26, 26 < BMI ≤ 29 or > 29, respectively.-One point was assigned for those that selected “Climbing one flight of stairs without resting” and/or “Walking 100 m” to the question: “Please tell me whether you have any difficulty doing each of the everyday activities on this card” fulfilling slowness criterion.-Participants that answered, “One to three times a month” or “hardly ever or never” to the question “How often do you engage in activities that require a low or moderate level of energy such as gardening, cleaning the car, or going for a walk?”, fulfilled the low activity criterion.

### 2.3. Polypharmacy

For this work, and given the data available, polypharmacy was defined, as the concurrent use of five or more medications per day. Polypharmacy was defined as previously described [14].

### 2.4. Exposure and Outcomes (All-Cause Mortality)

In case the participant was deceased between wave 6 and wave 7, an “end-of-life” interview was conducted with a proxy (family or household member, or any other person of the closer social network of the deceased participant). The month and year of the death were used.

### 2.5. Covariates

Sociodemographic characteristics such as age and gender were self-reported, and values from wave 6 (baseline) were used.

### 2.6. Analysis

To assess the prevalence of frailty status, polypharmacy, and mortality, we performed a descriptive analysis of the data collected, as a first approach. Three age groups were created (65–74, 75–84, 85+ years old), and the prevalence results were standardized by age and gender.

The timeline was created with the month and year of the interview of wave 6 as month 0, until the time of death or end of observation (month and year of the death or interview, in wave 7). Survival analysis, considering the competing risk and the event of interest, was performed to analyze participants’ survival. The event of interest was always death and the competing risk event was either polypharmacy and frailty (Figure 1, Figure 2 and Figure 3), a combination of polypharmacy with gender (Figure 4), or a combination of polypharmacy and age (Figure 5). Kaplan-Meier curves were used, and the log-rank test was used to evaluate the difference between groups.

Given the multilevel structure of data, with individuals nested in each country, we used a multilevel binary logistic regression approach, considering death as the dependent variable. We performed multilevel, univariable logistic regression, considering each covariate in turn, to identify factors potentially associated with the outcome variable. Significant covariates from this first step were included in a multilevel, multivariable logistic regression model. Age (65–74 years, 75–84 years, ≥85 years), gender (female, male) were used as covariates, as well as a combination of polypharmacy with frailty (non-frail non-polymedicated individuals, non-frail polymedicated individuals, pre-frail non-polymedicated individuals, pre-frail polymedicated individuals, frail non-polymedicated individuals, frail polymedicated individuals).

The country was entered as a random effect. The final model was composed only of significant covariates, selected using a backward selection method. The odds ratio and 95% confidence intervals (CI) were reported. The significance level was set at 0.05. All analyses were performed using the software IBM SPSS Statistics (Version 26.0. Armonk, NY, USA).

### 2.7. Ethics Approval

The SHARE study is subjected to continuous ethics review, and from wave 4 onwards, it was reviewed and approved by the Ethics Council of the Max Planck Society. This study required no ethics approval once it was performed using publicly available data.

## 3. Results

Of the 24,693 participants included in this study, the mean age was 74.5 ± 6.9 years old, and 55.5% were female (Table 1). The overall prevalence of frailty status was 40.4% non-frail (between 26.6% in Poland to 52.7% in Denmark), 47.3% pre-frail (between 38.7% in Austria and Denmark to 57.7% in Estonia) and 12.3% frail (between 4.2% in Switzerland to 20.8% in Israel) (Table 1). Increasing age was associated with higher rates of pre-frailty and frailty; 43.4% and 6.4% for those aged 65-74, 51.7% and 15.8% for those aged 75-84, and 52.4% and 32.9% for those aged 85 years or older, of pre-frailty and frailty, respectively. Pre-frailty and frailty were found to be more common among females, comparing with males (49.1% and 14.8% vs. 45.0% and 9.3%) (data not shown).

The overall prevalence of polypharmacy was 31.3% (between 22.9% in Slovenia to 45.6% in Poland). Although males and females reported similar rates of polypharmacy (31.8% in female vs. 30.7% in males), along with age the prevalence of polypharmacy also increased (25.9%, 36.8% and 40.9%, for those age 65–74, 75–84 and 85+, respectively) (data not shown). Within the polymedicated individuals, 53.1% took medication for high blood cholesterol, 76.6% for high blood pressure, 28.5% for coronary diseases, 31.6% for other heart diseases, 33.1% diabetes, 35.3% for joint pain, 26.4% for other pain, 19.2% for sleep problems, 13.6% for anxiety or depression, 12.1% for osteoporosis, 20.6% for stomach burns, 7.8% for chronic bronchitis, 6.5% for suppressing inflammation (only glucocorticoids or steroids), and 29.0% for other conditions.

Polypharmacy was found to be almost 3 times more prevalent in frail individuals and 2 times in pre-frail individuals, when compared with non-frail ones (59.4%, 39.4%, 18.4%, respectively) (Table 1). Within non-frail individuals, polypharmacy prevalence was higher among males comparing (19.7% vs. 17.2%, in females). Within pre-frail and frail individuals, polypharmacy prevalence was higher in females (56.3% vs. 43.7%, among pre-frail individuals, and 66.9% vs. 59.4%, among frail individuals).

Median follow-up time was 25 months (23–26, IQR) for those who stayed alive, and 14 months (8–19, IQR) for those who deceased. Throughout follow-up, 94.8% (23 399) of the participants remained alive while 5.2% (1294) of the participants died from all-cause of mortality. The average age at baseline of the individuals who deceased is 80.1 ± 7.8 years, comparing with 74.2 ± 6.7 years for those alive at the end of the 30 months follow-up (Table 2).

Among frail and pre-frail individuals, the mortality rates were higher, comparing with non-frail ones (15.6%, 5.3% and 2.0%, respectively), and within each group, even higher for polymedicated participants (Table 3). Survival analysis showed significant differences between non-frail non-polymedicated and polymedicated individuals (Log rank: χ^2^= 4.470, *p* = 0.035) (Figure 1) and between pre-frail non-polymedicated and polymedicated individuals (Figure 2) (Log rank: χ^2^= 18.281, *p* < 0.001). On the contrary, no significant differences were found between frail non-polymedicated and polymedicated individuals (Log rank: χ^2^= 0.074, *p* = 0.785) (Figure 3). Mortality rates were also higher in polymedicated individuals (7.8%), in male (6.7% vs. 4.1%, in female), in older participants (2.5%, 6.2%, 17.4%, for those age 65–74, 75–84 and 85+, respectively) (Table 3). Survival analysis showed significant differences between non-polymedicated and polymedicated individuals within gender (Figure 4) ((Log rank: χ^2^= 240.729, *p* < 0.001) and age (Figure 5) (Log rank: χ^2^= 1045.855, *p* < 0.001).

A significant association was found between mortality and gender, age, frailty status and polypharmacy, on both unadjusted and adjusted models (Table 4). Men were 2.1 times more prone to die within the 30 months follow up. The likelihood of death also increased with increasing age, comparing with those aged 65–74 years old ([OR = 2.051 (1.776–2.367)], for those aged 75–84 years old and [OR = 5.272 (4.477–6.207)], for those aged 85 or more years old). A significant association was also found between polypharmacy and frailty with mortality, except for the non-frail polymedicated individuals. Comparing with non-polymedicated non-frail individuals, polymedicated frail individuals are 7 times, non-polymedicated frail individuals are 6 times, polymedicated pre-frail individuals are 2.9 times and non-polymedicated pre-frail individuals are 2.1 times more prone to die with 30 months (Table 4).

## 4. Discussion

Polypharmacy and frailty are two very prevalent conditions in Europe and represent serious health issues that lead to lower quality of life, higher healthcare expenditures and a problem to society in general. Although polypharmacy is a universal and common term there is no universal definition [14,27]. There are numerical definitions that vary from two to 11 or more drugs taken, but there also descriptive definitions. The same happens to frailty as there are several tools to assess frailty since it is a multidimensional syndrome that includes a lot of variables. This makes it difficult to compare data from different studies of both conditions since it can lead to a wide range of results.

A previous study based on wave 6 of SHARE [14] showed a polypharmacy prevalence of 32.1%. It also showed that polypharmacy increases with age (26.7.1%, 37.5% and 43.1%, for those age 65–74, 75–84 and 85+, respectively) as well as it being more frequent in females than males (33.0% vs. 31.2%) [14], which agrees with the results of this work. Polypharmacy is known to be an independent predictor of mortality [26]. As the individual gets older, they are more likely to develop several diseases and comorbidities that lead to polypharmacy. With that being said, it is hard to determine whether polypharmacy is a cause of the increased risk of mortality or a consequence of ageing and multimorbidity. The intake of several drugs is associated with severe adverse drug reactions due to the different physiologic variations that are associated with ageing leading the body to be more sensitive to the medication, having a cumulative effect of several medications on the renal or hepatic systems that would lead to serious interactions [26,28]. With ageing, there is a decrease in liver mass, extending the half-life of several medications [29], and kidney’s function also starts to decline, which has an impact on the elimination of many medications that are excreted through the kidneys [30]. The risk for falls, fractures and aspiration pneumonia is increased with medications affecting the central nervous system since they might cause adverse effects such as sleepiness, reduced memory, confusion, among others [31]. Assessing polypharmacy in older people may be valuable to determine the mortality risk but deprescribing approaches need to be studied to better know the improvements in health risks of these individuals [26,32]. Chronic pathologies associated with ageing, such as arterial hypertension, diabetes mellitus, dyslipidaemia, osteoarticular degenerative pathology, chronic obstructive pulmonary disease, atrial fibrillation, and dementia, potentiate polypharmacy, with a consequent increase in healthcare costs, falls and hospitalizations rates, adverse drug reactions, drug-drug reactions, medication nonadherence and mortality [33,34,35]. In this study, within polymedicated individuals, cardiovascular diseases were the most common diseases, followed by diabetes, joint pain and other type of pain, sleep problems, anxiety, depression and osteoporosis, which are common conditions among the polymedicated older population [36]. A previous study on frailty status based on wave 6 from SHARE, showed a prevalence of pre-frailty and frailty of 42.9% and 7.7% respectively [27]. This study also showed that pre-frailty and frailty increase with age, which is in agreement with our study [27]. It also shows that pre-frailty and frailty are more common among females comparing to males (45.4% and 9.1% vs. 39.7% and 6.0%, respectively) which is also in line with our study (49.1% and 14.8% vs. 45.0% and 9.3%) [27]. In the present study, we found a higher prevalence of pre-frailty and frailty of 47.3% and 12.3% respectively, values slightly higher than those reported by Manfredi et al., which can be explained by the difference between the median age of both studies (67.45 ± 9.71 years vs. 74.5 ± 6.9 years). The development of frailty and its progression can have a lot of contributors and includes several aspects such as biological, sociodemographic, clinical and lifestyle [37]. Results from an overview of systematic reviews showed that 24/24 studies demonstrate an increased mortality risk for frail individuals while 7/7 showed a significant impact of pre-frailty in mortality [38]. The relation between frailty and mortality risk among other conditions in older adults was also studied, and it was found to increase the risk of death among diabetic patients [39], after hip fracture [40], after pelvic surgery [41], after mitral valve replacement [42], among others. Frailty is also associated with worse short term-term postoperative outcomes and mortality [43]. One of the risks associated with frailty appears to be multimorbidity. These different concepts may contribute to each other as a 2018 systematic review shows that most frail individuals have multimorbidity and a small fraction of multimorbid older people have frailty leading to an inconclusive association between the two, with a bidirectional association being the most plausible [37,44].

Studying the association between polypharmacy and frailty, we found that frail and pre-frail individuals have a higher prevalence of polypharmacy, with the pre-frail status being two times more polymedicated and frail status three times when comparing to polymedicated non-frail individuals. These results agree with previous publications [26,45,46,47,48]. A study from Singapore, using the Fried phenotype and the same definition of polypharmacy shows that (pre-)frail individuals are more prone to polypharmacy than robust individuals (41.5%, 29.8% and 18.1 in frail, pre-frail and robust, respectively) [20]. Besides this, it showed that for polymedicated individuals the mortality rates increase with the number of drugs, and the frail individuals showed even higher mortality rates. Moreover, the combination of frailty and excessive polypharmacy (polypharmacy being > 5 drugs and excessive polypharmacy > 10 or more) lead to a risk of death by 6 times higher during the follow-up of 39 months [26]. Another Spanish study has similar conclusions regarding this topic [24]. When analysing both polymedicated and frail individuals our results are more in line with the French study. Concerning the polymedicated non-frail individuals, by the Spanish study, we found no significant association with mortality [OR = 1,289 (0.928–1.791), *p* = 0.130].

Kaplan-Meier curves for overall survival, (Figure 1 and Figure 2) showed that the cohort of participants with polypharmacy is at an increased risk of death (log-rank *p* < 0.05), except for of the frail polymedicated individuals when comparing to frail non-polymedicated individuals. Higher mortality risk within polymedicated participants along with age and for male agrees with previous studies [32,49]. Within frailty status, differences between polymedicated and non-polymedicated individuals also arose [2,26]. Multimorbidity associated with other factors may lead to polypharmacy and frailty which in turn may lead to a higher risk of mortality. This association further accentuates the need to have more consciousness regarding these conditions. With the high impact of polypharmacy on the health outcomes of patients, especially frail individuals, the potential harms of polypharmacy must be present for each new drug added to the patient’s regimen should be well thought and review the current drugs. When reviewing the drug and deprescribing it should be considered the comprehensive health and medication list to reduce potentially inappropriate medications and pursue more personalized and individualized therapeutic plans for each patient. Life expectancy and care goals need to be taken into account when reviewing a patient’s list with each medication’s benefits and problems being weighed [30]. Suitable monitoring procedures must be applied including necessary laboratory testing and patient education about monitoring themselves for possible adverse events and what to do when it happens. It is important to mention that it might be hard to determine whether a higher risk of mortality and adverse outcomes are directly related to the numbers of drugs taken or the multiple chronic conditions that the individual might have. Therefore, interventions like deprescribing should be done with caution, as more studies need to be done to show how it affects long-term outcomes such as mortality [17,26].

Limitations in this study are the fact that all data on SHARE were self-reported, which might raise questions about the reliability of the data, as people might not recall correctly all the information they are being asked. People included in this type of research surveys are volunteers, who are asked if they want to participate. Healthier people might be more inclined to participate while a high number of older people with comorbidities might decline, which might reduce the number of people with disabilities in this study. Also, this survey is self-reported so the veracity of the answers might be questionable. The research based on this database is limited to the questions that exist in it since we cannot change them to fit the type of research we want to do. Strengths are the high number of participants and the inclusion of 18 countries giving a close overview of these conditions in Europe. SHARE has some advantages when it comes to its survey since it is the same survey in each country with a direct translation. This makes for a strong comparison since we are comparing the same questions between countries. Other surveys like Eurostat also cover a wide range of countries but in each country the survey is different and adapted to the cultural differences of each country, making it difficult to make a comparison between them or have a reliable representation of the European people. Usually, it is different to compare studies and countries since they use different scales but since this study uses the same scale for frailty and polypharmacy, it allows us to make a comparison between those countries.

## 5. Conclusions

With this work, and through the use of a cohort of SHARE participants, it was possible to understand the trajectory of almost 25,000 people aged 65 or over, in Europe. This work reinforces the high prevalence of polypharmacy and frailty in the older population and clearly shows the prevalence of polypharmacy in non-frail, pre-frail and frail individuals, in 17 European countries, and Israel. At the same time, we were able to understand the 30-month mortality rates for the different groups. It becomes clear and evident that interventions are needed to decrease polypharmacy (especially if not necessary), and frailty, since it will impact society, health systems, caregivers, and older people, improving their quality of life, well-being, functional independence and their autonomy.

## Figures and Tables

**Figure 1 ijerph-18-03580-f001:**
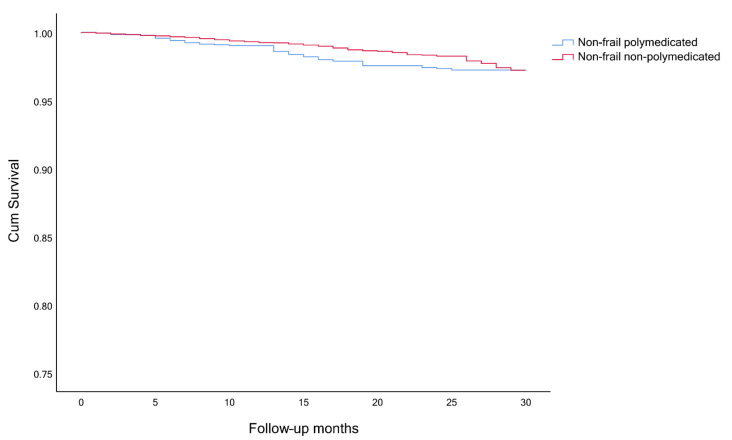
Survival curves for non-frail non-polymedicated and polymedicated individuals (Log rank: *p* = 0.035).

**Figure 2 ijerph-18-03580-f002:**
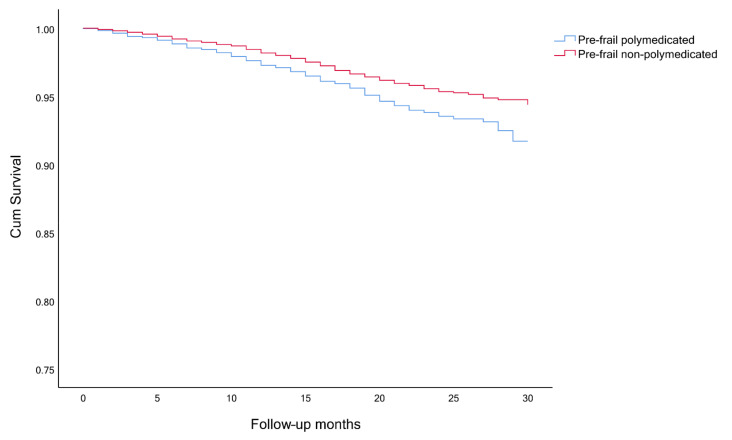
Survival curves for pre-frail non-polymedicated and polymedicated individuals (Log rank: *p* < 0.001).

**Figure 3 ijerph-18-03580-f003:**
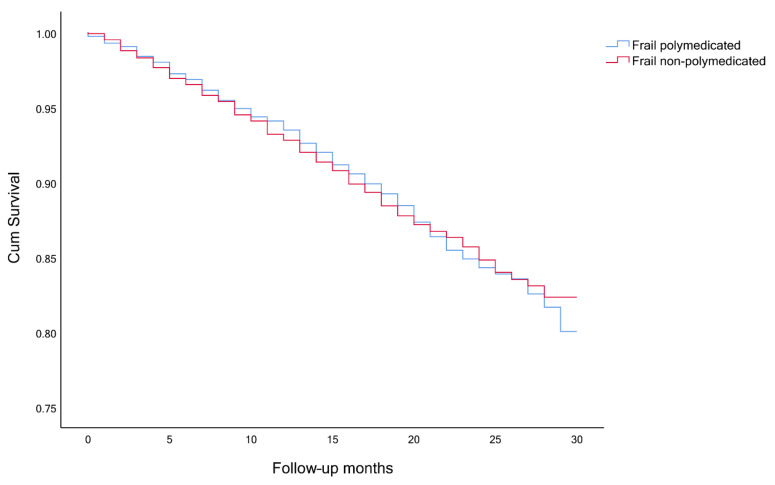
Survival curves for frail non-polymedicated and polymedicated individuals (Log rank: *p* = 0.785).

**Figure 4 ijerph-18-03580-f004:**
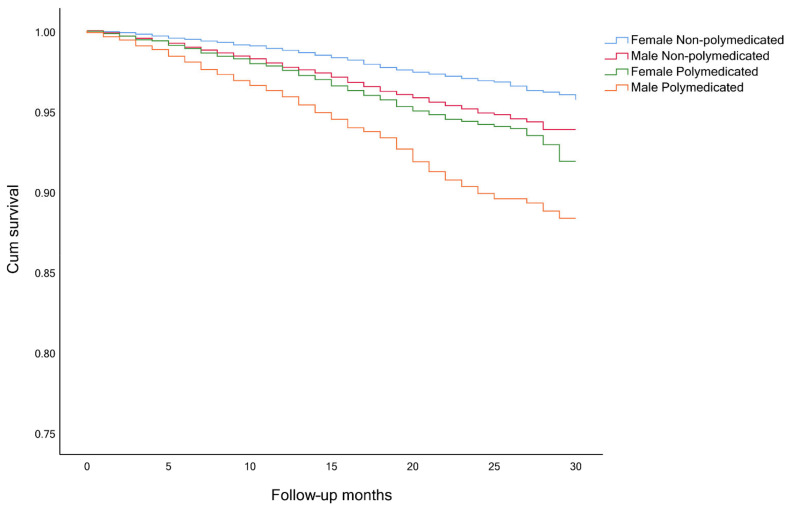
Survival curves non-polymedicated and polymedicated individuals split by gender (Log rank: *p* < 0.001).

**Figure 5 ijerph-18-03580-f005:**
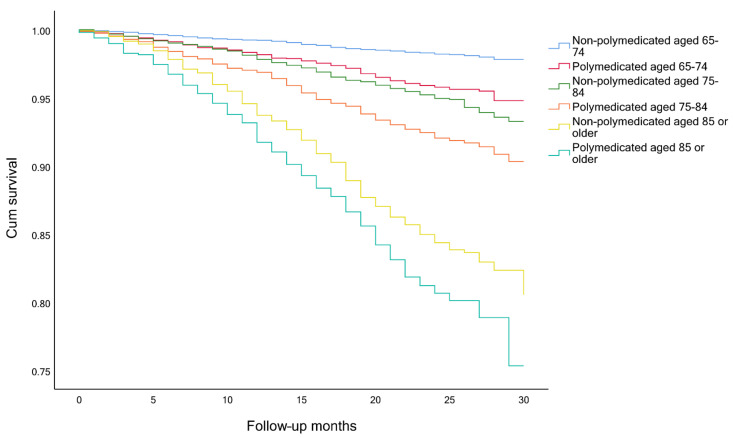
Survival curves non-polymedicated and polymedicated individuals split by age groups (Log rank: *p* < 0.001).

**Table 1 ijerph-18-03580-t001:** Detailed characteristics of the population included in the study, at baseline (wave 6–2015).

Country	Population N	Age Mean ± sd	Female %	Polypharmacy %	Frailty Status					
					Non-Frail		Pre-Frail		Frail	
					% Overall	% Polymedicated	% Overall	% Polymedicated	% Overall	% Polymedicated
Austria	1384	74.6 ± 6.7	58.7	30.0	50.1	16.0	38.7	39.8	11.3	58.3
Belgium	2065	74.6 ± 7.2	55.3	34.5	41.0	21.0	45.6	38.9	13.4	61.0
Croatia	770	73.1 ± 6.2	55.8	28.7	35.5	18.7	49.4	27.1	15.2	57.3
Czech Republic	2209	73.5 ± 6.2	58.0	40.2	46.9	27.1	44.2	47.7	8.9	71.9
Denmark	1207	74.3 ± 7.3	53.8	35.6	52.7	24.7	38.7	43.3	8.6	68.3
Estonia	2251	75.3 ± 6.7	65.3	28.2	28.5	16.7	57.7	29.2	13.9	47.4
France	1412	75.6 ± 7.6	58.4	33.5	34.6	15.6	50.1	36.7	15.3	63.4
Germany	1549	73.8 ± 6.4	48.8	28.6	49.9	17.5	43.0	36.3	7.1	60.0
Greece	1422	74.4 ± 6.6	52.6	23.5	38.0	12.4	49.4	24.5	12.6	53.1
Israel	742	74.9 ± 7.1	54.3	44.9	30.2	27.2	49.1	45.3	20.8	69.5
Italy	1817	73.9 ± 6.3	52.1	27.4	33.2	13.2	50.5	28.2	16.2	53.6
Luxembourg	404	72.9 ± 6.4	48.8	32.2	49.3	21.6	43.6	38.1	7.2	69.0
Poland	654	73.8 ± 6.9	57.0	45.6	26.6	29.3	54.1	46.3	19.3	65.9
Portugal	203	74.2 ± 6.6	56.2	44.3	33.0	25.4	46.3	41.5	20.7	81.0
Slovenia	1630	74.3 ± 6.7	57.2	22.9	47.1	11.3	41.8	29.2	11.1	48.6
Spain	2160	75.8 ± 7.2	53.9	29.1	29.1	12.9	52.1	27.6	18.8	58.5
Sweden	1715	75.1 ± 7.0	52.4	32.9	48.3	21.0	45.9	40.7	5.8	71.0
Switzerland	1099	74.9 ± 6.9	51.5	23.5	50.6	14.9	45.2	29.8	4.2	58.7
Total	24,693	74.5 ± 6.9	55.5	31.3	40.4	18.4	47.3	34.9	12.3	59.4

**Table 2 ijerph-18-03580-t002:** Characteristics, at baseline, for alive (IA) and deceased (ID) individuals, at the end of the follow-up period.

Country	NIA	NID	Age IA Mean ± sd	Age ID Mean ± sd	% Female IA	% FemaleID	% IA Polypharmacy	% ID Polypharmacy	Frailty Status					
									Non-Frail		Pre-Frail		Frail	
									% IA	% ID	% IA	% ID	% IA	% ID
Austria	1327	57	74.3 ± 6.5	80.1 ± 8.3	59.0	50.9	28.9	56.1	51.8	10.5	38.7	36.8	9.5	52.6
Belgium	1978	87	74.3 ± 7.1	81.3 ± 7.9	55.6	47.1	33.7	52.9	42.2	12.6	45.9	39.1	11.9	48.3
Croatia	714	56	72.7 ± 5.9	78.2 ± 8.0	57.6	33.9	28.4	32.1	37.1	14.3	49.3	50.0	13.6	35.7
Czech Republic	2098	111	73.3 ± 6.1	76.9 ± 7.3	58.8	43.2	39.2	58.6	48.1	23.4	43.9	50.5	8.0	26.1
Denmark	1127	80	73.8 ± 6.9	81.9 ± 9.1	53.8	53.8	34.3	53.8	55.1	18.8	38.6	40.0	6.3	41.3
Estonia	2118	133	75.0 ± 6.6	79.2 ± 7.5	66.4	47.4	27.9	32.3	29.4	14.3	58.3	48.1	12.4	37.6
France	1334	78	75.2 ± 7.4	82.4 ± 7.7	59.0	48.7	31.9	61.5	36.0	10.3	50.6	42.3	13.4	47.4
Germany	1499	50	73.6 ± 6.3	79.7 ± 7.5	49.2	36.0	28.1	44.0	50.9	20.0	42.8	48.0	6.3	32.0
Greece	1317	105	73.8 ± 6.4	81.4 ± 5.9	53.5	41.9	22.6	34.3	39.7	16.2	48.8	57.1	11.5	26.7
Israel	711	31	74.5 ± 6.8	83.7 ± 7.1	54.6	48.4	44.2	61.3	31.4	3.2	49.2	45.2	19.4	51.6
Italy	1727	90	73.6 ± 6.1	78.4 ± 7.0	53.0	35.6	26.6	42.2	34.2	14.4	50.4	52.2	15.3	33.3
Luxembourg	392	12	72.8 ± 6.3	75.3 ± 7.3	49.2	33.3	31.1	66.7	50.3	16.7	43.9	33.3	5.9	50.0
Poland	614	40	73.6 ± 6.9	76.8 ± 6.9	57.3	52.5	44.8	57.5	27.9	7.5	54.6	47.5	17.6	45.0
Portugal	192	11	73.8 ± 6.3	82.5 ± 7.4	57.8	27.3	43.2	63.6	34.9	0.0	46.4	45.5	18.8	54.5
Slovenia	1550	80	74.0 ± 6.6	79.9 ± 7.0	58.0	42.5	22.3	35.0	48.2	26.3	41.5	47.5	10.3	26.3
Spain	1998	162	75.4 ± 7.0	80.4 ± 7.8	55.1	40.1	28.1	42.0	30.3	14.2	52.7	44.4	17.0	41.4
Sweden	1643	72	74.8 ± 6.7	82.5 ± 8.8	52.9	41.7	32.2	50.0	49.5	20.8	45.4	56.9	5.1	22.2
Switzerland	1060	39	74.7 ± 6.8	80.6 ± 7.2	52.0	38.5	22.4	53.8	52.0	12.8	44.7	59.0	3.3	28.2
Total	23,399	1294	74.2 ± 6.7	80.1 ± 7.8	56.2	43.4	30.4	46.4	41.8	15.7	47.3	47.5	11.0	36.8

**Table 3 ijerph-18-03580-t003:** Detailed mortality rates within different groups (by gender, age, polymedicated, frailty status and polymedicated individuals by frailty status).

Country	Overall %	Gender		Age			Polypharmacy %	Frailty Status					
		Male %	Female %	65–74 %	75–84 %	85+ %		Non-Frail		Pre-Frail		Frail	
								Overall %	Polymedicated %	Overall %	Polymedicated %	Overall %	Polymedicated %
Austria	4.1	4.9	3.6	2.0	5.0	12.7	7.7	0.9	1.8	3.9	5.6	19.2	19.8
Belgium	4.2	5.0	3.6	1.9	4.5	14.8	6.5	1.3	0.0	3.6	5.8	15.2	12.1
Croatia	7.3	10.9	4.4	4.6	8.6	30.2	8.1	2.9	1.7	7.4	6.9	17.1	15.5
Czech Republic	5.0	6.8	3.7	3.3	6.4	16.0	7.3	2.5	7.4	5.7	6.8	14.8	17.2
Denmark	6.6	6.6	6.6	2.8	7.6	24.4	10.0	2.4	5.0	6.9	6.6	31.7	10.8
Estonia	5.9	9.0	4.3	3.5	6.0	16.5	6.8	3.0	3.9	4.9	6.9	16.0	19.7
France	5.5	6.8	4.6	1.8	5.9	17.2	10.1	1.6	2.5	4.7	6.9	17.1	35.2
Germany	3.2	4.0	2.4	1.3	4.2	13.5	5.0	1.3	4.5	3.6	9.3	14.5	17.9
Greece	7.4	9.1	5.9	1.8	9.7	32.5	10.8	3.1	0.0	8.5	10.8	15.6	18.5
Israel	4.2	4.7	3.7	0.5	4.5	19.8	5.7	0.4	1.1	3.8	4.9	10.4	15.4
Italy	5.0	6.7	3.4	2.6	6.7	15.4	7.6	2.2	0.0	5.1	4.2	10.2	11.2
Luxembourg	3.0	3.9	2.0	2.2	4.0	7.4	6.2	1.0	3.9	2.3	6.4	20.7	17.0
Poland	6.1	6.8	5.6	3.7	8.8	12.3	7.7	1.7	5.9	5.4	5.5	14.3	13.3
Portugal	5.4	9.0	2.6	0.8	9.5	18.2	7.8	0.0	2.3	5.3	4.5	14.3	20.0
Slovenia	4.9	6.6	3.6	2.2	6.6	15.9	7.5	2.7	0.0	5.6	7.7	11.6	11.8
Spain	7.5	9.7	5.6	3.8	8.0	18.9	10.8	3.7	2.3	6.4	8.5	16.5	10.2
Sweden	4.2	5.1	3.3	1.7	3.7	17.5	6.4	1.8	3.7	5.2	5.5	16.0	12.2
Switzerland	3.5	4.5	2.7	1.2	5.1	9.9	8.1	0.9	2.0	4.6	6.8	23.9	14.9
Total	5.2	6.7	4.1	2.5	6.2	17.4	7.8	2.0	2.7	5.3	6.5	15.6	15.8

**Table 4 ijerph-18-03580-t004:** Association of age, gender, frailty status and polypharmacy with mortality: unadjusted and adjusted models.

			Unadjusted Model			Adjusted Model		
	N	N Deceased (%)	OR	CI 95	*p*	OR	CI 95	*p*
Age								
65–74 years	13,451	330 (2.4)	1	-	-	1	-	-
75–84 years	8845	548 (6.2)	2.630	2.287–3.025	<0.001	2.051	1.776–2.367	<0.001
≥85 years	2397	416 (17.3)	8.487	7.279–9.896	<0.001	5.272	4.477–6.207	<0.001
Gender								
Female	13,709	564 (4.1)	1	-	-	1	-	-
Male	10,984	732 (6.7)	1.692	1.511–1.895	<0.001	2.134	1.894–2.405	<0.001
Frailty Status vs. Polypharmacy								
Non-frail non-polymedicated	8134	154 (1.9)	1	-	-	1	-	-
Non-frail polymedicated	1840	49 (2.7)	1.423	1.027–1.971	0.034	1.289	0.928–1.791	0.130
Pre-frail non-polymedicated	7598	350 (4.6)	2.472	2.038–2.999	<0.001	2.091	1.718–2.546	<0.001
Pre-frail polymedicated	4074	265 (6.5)	3.633	2.966–4.451	<0.001	2.874	2.335–3.539	<0.001
Frail non-polymedicated	1236	189 (15.3)	9.252	7.392–11.582	<0.001	5.973	4.711–7.573	<0.001
Frail polymedicated	1811	287 (15.8)	9.754	7.944–11.975	<0.001	6.981	5.624–8.664	<0.001

## Data Availability

The data presented in this study are openly available in http://www.share-project.org/ (SHARE Wave 6 - DOI: 10.6103/SHARE.w6.710; SHARE Wave 7 - DOI: 10.6103/SHARE.w7.711)

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
