# Peer review of "Frailty Status and Polypharmacy Predict All-Cause Mortality in Community Dwelling Older Adults in Europe"

_ijerph, 2021, doi:10.3390/ijerph18073580_

Round 1
Reviewer 1 Report
It is my pleasure to review your valuable manuscript.
- In the title part, “all-causes mortality” should fix “all-cause mortality”. Please reconsider and revise this part.
- Polypharmacy defines only by the numbers of oral medicine. I am considering the purpose of medications also important. So, you should add the main medical history which became the cause of medications. Please reconsider and revise those.
- P2 Line 81 2.2 Frailty: You explained the definition of frailty. But you only wrote, “Frailty status was defined as previously described [27]”. You should describe in detail the differences among non, pre-frailty, and frailty in this part. Because it is an important concept in this manuscript. Please reconsider and revise those.
- P3 Line 102 You mentioned that “Timeline was created with month and year of interview of wave 6 as month 0,” I cannot understand the meaning of wave 6 and 7 before this part. So, please reconsider and revise this issue.
- This study aims to clarify the impact of frailty and polypharmacy on mortality rates. So, you should adjust the age factor for this purpose. In the statistical results, it is clear that higher age has relations to higher mortality rates. Frailty rates also have relations to a higher age. So, please reconsider and revise these issues.
- According to revisions of your results, you should reconsider and revise the discussion part.
Author Response
Response to Reviewer 1 Comments
Point 1: In the title part, “all-causes mortality” should fix “all-cause mortality”. Please reconsider and revise this part.
Response 1: All “all-causes” were changed to “all-cause”.
Point 2: Polypharmacy defines only by the numbers of oral medicine. I am considering the purpose of medications also important. So, you should add the main medical history which became the cause of medications. Please reconsider and revise those.
Response 2: We would like to thank the reviewer for this comment. It would make sense to demonstrate what are the main diseases that led the participants to be polymedicated. Thus, we added a sentence that demonstrates which diseases are more prevalent in this population, and what percentage of individuals took medication for each disease.
“Within the polymedicated individuals, 53.1% took medication for high blood cholesterol, 76.6% for high blood pressure, 28.5% for coronary diseases, 31.6% for other heart diseases, 33.1% diabetes, 35.3% for joint pain, 26.4% for other pain, 19.2% for sleep problems, 13.6% for anxiety or depression, 12.1% for osteoporosis, 20.6% for stomach burns, 7.8% for chronic bronchitis, 6.5% for suppressing inflammation (only glucocorticoids or steroids), and 29.0% for other conditions.” (lines 171-176)
Point 3: P2 Line 81 2.2 Frailty: You explained the definition of frailty. But you only wrote, “Frailty status was defined as previously described [27]”. You should describe in detail the differences among non, pre-frailty, and frailty in this part. Because it is an important concept in this manuscript. Please reconsider and revise those.
Response 3: This comment is very pertinent, and we would like to thank you for it. We added the information regarding the classification of the frailty status.
“….. and that is based on five dimensions: exhaustion, shrinking, weakness, slowness, and low activity.
For each fulfilled criterion, one point was assigned to each participant. Individuals with zero points were classified as robust, those with 1 or 2 points as pre-frail and those with 3 to 5 points were classified as frail.
- One point was assigned for those with a positive answer to “In the last month, have you had too little energy to do the things you wanted to do?”, fulfilling the exhaustion criterion.
- Those that reported “diminution in desire for food” to “What has your appetite been like in the last month?” or, in the case of an encodable answer to this question, those that reported “less” to “So, have you been eating more or less than usual?”, had one point in the shrinking criterion.
- Weakness criteria, derived from the highest of 4 handgrip strength measurements, adjusted by gender and body mass index (BMI), and was fulfilled by men with handgrip strength ≤ 29, ≤ 30 or ≤ 32, if BMI ≤ 24, 24 < BMI ≤28 or ≤ 32, respectively; and by women with handgrip strength ≤ 17, ≤3, ≤ 18 or ≤ 21 if BMI ≤ 23, 23 < BMI ≤26, 26 < BMI ≤ 29 or > 29, respectively.
- One point was assigned for those that selected “Climbing one flight of stairs without resting” and/or “Walking 100 meters” to the question: “Please tell me whether you have any difficulty doing each of the everyday activities on this card” fulfilling slowness criterion.
Participants that answered, “One to three times a month” or “hardly ever or never” to the question “How often do you engage in activities that require a low or moderate level of energy such as gardening, cleaning the car, or going for a walk?”, fulfilled the low activity criterion.” (lines 88-112)
Point 4: P3 Line 102 You mentioned that “Timeline was created with month and year of interview of wave 6 as month 0,” I cannot understand the meaning of wave 6 and 7 before this part. So, please reconsider and revise this issue.
Response 4: Thanks for point out this flaw. We added a sentence on the methods section, hoping that it will elucidate better the reader of what is Wave 6 and Wave 7.
” Since its inception in 2004, SHARE has released data every 2 years, having already released 7 waves of data. Wave 6 (with data from 2015) and wave 7 (with data from 2017), are the most recent waves. In this work, we used data from wave 6 and 7.” (lines 77-79)
Point 5: This study aims to clarify the impact of frailty and polypharmacy on mortality rates. So, you should adjust the age factor for this purpose. In the statistical results, it is clear that higher age has relations to higher mortality rates. Frailty rates also have relations to a higher age. So, please reconsider and revise these issues.
Response 5: First of all, we would like to thank the reviewer for this comment. In fact, in the statistical analysis, (table 4), we present the unadjusted model (univariate analysis) and the adjusted model (multivariate analysis). Using multivariable methods, we were able to estimate the association between the outcome (mortality) considering the impact of more exposure (age, gender, polypharmacy, and frailty). In essence, this multiple variable analysis allowed us to assess the independent effect of each of the exposures. Therefore, we believe that our results are already adjusted by the age factor.
Point 6: According to revisions of your results, you should reconsider and revise the discussion part.
Response 6: Thank you for this comment. Considering your inputs, we were able to improve the quality of our manuscript. Changes on the discussion and conclusion were made. (lines 268-271; 354-362)
Reviewer 2 Report
The authors have done a nice job assessing the mortality risk in frailty and polypharmacy. The article needs to be improved before could be published.
The article needs to be edited as there are several grammatical errors especially in regard with the prepositions.
In the methods the definition of frailty has be referred to a publication. Considering the fact that is the basis of this article, I suggest the definition and how pre-frailty is defined to be added to the article.
The difference of the prevalence of frailty and polypharmacy among the two genders is missing. The interaction between frailty and polypharmacy is also not clarified.
The conclusion needs to be improved. The implications and how these data could be used are missing.
Author Response
Response to Reviewer 2 Comment
Point 1: The article needs to be edited as there are several grammatical errors especially in regard with the prepositions.
Response 1: We would like to thank the reviewer for this comment, which allowed us to improve the quality of our work. Corrections were made throughout the article, and we corrected the misuse of prepositions (lines 27, 33, 38, 58, 59, 63, 65, 84, 103….)
Point 2: In the methods the definition of frailty has be referred to a publication. Considering the fact that is the basis of this article, I suggest the definition and how pre-frailty is defined to be added to the article.
Response 2: This comment is very pertinent, and we would like to thank you for it. We added the information regarding the classification of the frailty status.
“….. and that is based on five dimensions: exhaustion, shrinking, weakness, slowness, and low activity.
For each fulfilled criterion, one point was assigned to each participant. Individuals with zero points were classified as robust, those with 1 or 2 points as pre-frail and those with 3 to 5 points were classified as frail.
- One point was assigned for those with a positive answer to “In the last month, have you had too little energy to do the things you wanted to do?”, fulfilling the exhaustion criterion.
- Those that reported “diminution in desire for food” to “What has your appetite been like in the last month?” or, in the case of an encodable answer to this question, those that reported “less” to “So, have you been eating more or less than usual?”, had one point in the shrinking criterion.
- Weakness criteria, derived from the highest of 4 handgrip strength measurements, adjusted by gender and body mass index (BMI), and was fulfilled by men with handgrip strength ≤ 29, ≤ 30 or ≤ 32, if BMI ≤ 24, 24 < BMI ≤28 or ≤ 32, respectively; and by women with handgrip strength ≤ 17, ≤3, ≤ 18 or ≤ 21 if BMI ≤ 23, 23 < BMI ≤26, 26 < BMI ≤ 29 or > 29, respectively.
- One point was assigned for those that selected “Climbing one flight of stairs without resting” and/or “Walking 100 meters” to the question: “Please tell me whether you have any difficulty doing each of the everyday activities on this card” fulfilling slowness criterion.
- Participants that answered, “One to three times a month” or “hardly ever or never” to the question “How often do you engage in activities that require a low or moderate level of energy such as gardening, cleaning the car, or going for a walk?”, fulfilled the low activity criterion.” (lines 88-112)
Point 3: The difference of the prevalence of frailty and polypharmacy among the two genders is missing.
Response 3: Thanks for pointing out this issue. We have now added that information. We had to restructure the paragraph, and we added the sentence: “Within non-frail individuals, polypharmacy prevalence was higher among males comparing (19.7% vs. 17.2%, in females). Within pre-frail and frail individuals, polypharmacy prevalence was higher in females (56.3% vs 43.7%, among pre-frail individuals, and 66.9% vs. 59.4%, among frail individuals).” (lines 179-182)
Point 4: The interaction between frailty and polypharmacy is also not clarified.
Response 4: We would like to thank you for this comment. In fact, as mentioned in the introduction, the relationship between frailty and polypharmacy has already been studied in other articles [1-4]. Polypharmacy increases the risk of people suffering from frailty, and frailty also increases polypharmacy. However, the relationship between these two conditions is not yet fully understood, and research has been done to understand this. Bearing this in mind, with this work we did not intended to understand the relationship between the conditions, but the consequences of the combination of both on mortality rates.
Point 5: The conclusion needs to be improved. The implications and how these data could be used are missing.
Response 5: We would like to thank again the reviewer for this comment, which allowed us to improve the quality of our work. We changed and enhanced the conclusion, as shown below:
“With this work, and through the use of a cohort of SHARE participants, it was possible to understand the trajectory of almost 25000 people aged 65 or over, in Europe. This work reinforces the high prevalence of polypharmacy and frailty in the older population and clearly shows the prevalence of polypharmacy in non-frail, pre-frail and frail individuals, in 17 European countries, and Israel. At the same time, we were able to understand the 30-month mortality rates for the different groups. It becomes clear and evident that interventions are needed to decrease polypharmacy (especially if not necessary), and frailty, since it will impact society, health systems, caregivers, and older people, improving their quality of life, well-being, functional independence and their autonomy.” (lines 354-361).
- Veronese, N., et al., Polypharmacy Is Associated With Higher Frailty Risk in Older People: An 8-Year Longitudinal Cohort Study. J Am Med Dir Assoc, 2017. 18(7): p. 624-8.
- Saum, K.U., et al., Is Polypharmacy Associated with Frailty in Older People? Results From the ESTHER Cohort Study. J Am Geriatr Soc, 2017. 65(2): p. e27-e32.
- Yuki, A., et al., Polypharmacy is associated with frailty in Japanese community-dwelling older adults. Geriatr Gerontol Int, 2018. 18(10): p. 1497-1500.
- Gutiérrez‐Valencia, M., et al., The relationship between frailty and polypharmacy in older people: A systematic review, in Br J Clin Pharmacol. 2018. p. 1432-44.
Round 2
Reviewer 1 Report
Thank you for your wonderful effort to revise the manuscript. I have no additional comments for it. Thank you again.